# Bmp8a deletion leads to obesity through regulation of lipid metabolism and adipocyte differentiation

Shenjie Zhong [1,2], Lihui Chen[1,2], Xinyi Li[1,2], Xinyuan Wang[1,2], Guangdong Ji[1,2], Chen Sun [1,2✉] & Zhenhui Liu [1,2✉]

The role of bone morphogenetic proteins (BMPs) in regulating adipose has recently become a field of interest. However, the underlying mechanism of this effect has not been elucidated. Here we show that the anti-fat effect of Bmp8a is mediated by promoting fatty acid oxidation and inhibiting adipocyte differentiation. Knocking out the *bmp8a* gene in zebrafish results in weight gain, fatty liver, and increased fat production. The *bmp8a*$^{-/-}$ zebrafish exhibits decreased phosphorylation levels of AMPK and ACC in the liver and adipose tissues, indicating reduced fatty acid oxidation. Also, Bmp8a inhibits the differentiation of 3T3-L1 pre-adipocytes into mature adipocytes by activating the Smad2/3 signaling pathway, in which Smad2/3 binds to the central adipogenic factor PPARγ promoter to inhibit its transcription. In addition, lentivirus-mediated overexpression of Bmp8a in 3T3-L1 cells significantly increases NOD-like receptor, TNF, and NF-κB signaling pathways. Furthermore, NF-κB interacts with PPARγ, blocking PPARγ's activation of its target gene *Fabp4*, thereby inhibiting adipocyte differentiation. These data bring a signal bridge between immune regulation and adipocyte differentiation. Collectively, our findings indicate that Bmp8a plays a critical role in regulating lipid metabolism and adipogenesis, potentially providing a therapeutic approach for obesity and its comorbidities.

[1] College of Marine Life Science and Institute of Evolution & Marine Biodiversity, Ocean University of China, Qingdao 266003, China. [2] Laboratory for Marine Biology and Biotechnology, Pilot National Laboratory for Marine Science and Technology (Qingdao), Qingdao 266003, China. ✉email: sunchen@ouc.edu.cn; zhenhuiliu@ouc.edu.cn

Obesity and overweight have become a worldwide epidemic. Obesity is strongly associated with many metabolic and cardiovascular diseases, such as type 2 diabetes, dyslipidemia, hypertension, some types of cancer, and osteoarthritis[1–3]. Generally, obesity is characterized by a massive expansion of white adipose tissue due to an increase in the size or number of adipocytes and a decrease in lipolysis. Therefore, identifying the factors that regulate adipose tissue expansion and elucidating the mechanisms are vital for public health and will help formulate therapeutic strategies and targets for the treatment of obesity and its associated comorbidities.

Adipogenesis is regulated by a variety of signaling pathways[4–7], and bone morphogenetic proteins (BMPs) are a relatively recent addition to the adipose regulation field. BMP belongs to the transforming growth factor-β (TGF-β) superfamily, which is highly conserved in developing vertebrates ranging from humans to zebrafish[8,9]. They were initially discovered as inducers of bone and cartilage[10], but have been known to be critical in morphogenetic activities and cell differentiation throughout the body, including the development of adipose tissue and adipogenic differentiation[11–15]. BMP2, BMP4, and BMP6 have been shown to promote white adipogenesis in mesenchymal stem cells[16–18]. BMP7 and BMP8B can induce the expression of UCP1, a marker gene of brown adipocyte, and promote brown adipogenesis or enhance the thermogenesis of brown adipose tissue[19–23]. In addition, BMP3B suppresses adipogenesis of 3T3-L1 cells[24,25]. Overexpression of BMP9 in the mouse liver significantly alleviates hepatic steatosis and obesity-related metabolic syndrome[26].

BMP8A is almost absent from brown adipose tissue, whereas it is enriched in white adipose tissue[23]. Our previous studies have shown that Bmp8a can accelerate the uptake of yolk sac in zebrafish at 3-dpf (days post fertilization), indicating that Bmp8a may play a key role in fat metabolism[27]. Nevertheless, the overall metabolic regulatory function of BMP8A is not fully understood. Here, we report that bmp8a[-/-] zebrafish display obesity and fatty liver. Deletion of bmp8a in zebrafish leads to the accumulation of liver TG by downregulating phosphorylation of AMP-activated protein kinase (AMPK) and acetyl-CoA carboxylase (ACC). Furthermore, Bmp8a inhibits the differentiation of 3T3-L1 pre-adipocytes into mature adipocytes through the Smad2/3 pathway. Interestingly, we also found that the interaction of NF-κB and PPARγ mediates the effect of Bmp8a on adipogenesis, providing a signaling bridge between immunomodulatory and adipocyte differentiation. We present a previously unrecognized insight into Bmp8a-mediated adipogenesis.

## Results

**Weight gain, fatty liver and increased fat production in bmp8a[-/-] zebrafish.** We have found that recombinant Bmp8a protein is able to accelerate the absorption of yolk sac in 3 dpf zebrafish, indicating a function of Bmp8a in regulating lipid metabolic processes[27]. Here the bmp8a[-/-] zebrafish was used to perform further analysis[28]. We monitored the body weight of wild type (WT) and bmp8a[-/-] zebrafish fed with a high-fat diet (HFD) and found that the body weight of bmp8a[-/-] zebrafish was gradually higher than WT zebrafish (Fig. 1a, b). Next, we calculated the body weight of male and female zebrafish separately, which bmp8a[-/-] zebrafish had a significant increase in body weight compared to wild-type zebrafish, independent of sex (Fig. 1c). Furthermore, bmp8a deficiency in zebrafish induced hyperplastic morphology of visceral adipose tissue (VAT) (Fig. 1d). Oil Red O staining on the liver sections showed prominent fatty liver in bmp8a[-/-] zebrafish (Fig. 1e), which was confirmed by analysis of TG and TC levels in liver tissue (Fig. 1f, g). Zebrafish yolk sac is a quantifiable limited energy source mainly consumed during the first week of larval development and has unique advantages in detecting changes in body lipid metabolism[29]. All changes in fat content could be relatively visually quantified by Nile red fluorescence microscopy of live zebrafish larvae. We found that bmp8a deletion resulted in a detectable increase in fat, in zebrafish at 3- and 7-day larvae (Fig. 1h-k). Importantly, mutation of bmp8a in zebrafish leads to increased lipid droplets in viscera and other sites (Fig. 1l). Taken together, HFD-induced obesity and hepatic steatosis are more severe in bmp8a[-/-] zebrafish than WT zebrafish.

**Impaired glucose and fat metabolism in bmp8a[-/-] zebrafish.** We further found that HFD-fed bmp8a[-/-] zebrafish had higher levels of blood GLU, TG, and TC than WT zebrafish (Fig. 2a). To determine the molecular basis of the metabolic changes in bmp8a[-/-] zebrafish, we performed gene expression analyses by Quantitative real-time PCR (qRT-PCR) technology. It was shown that in the liver or adipose tissue of bmp8a[-/-] zebrafish fed a high-fat diet, genes for lipolytic enzymes (lpl and lipc), insulin-sensitizing hormone (adiponectin), transcription activators and coactivators that induce fat metabolism (pgc-1α and pparα), mitochondrial proteins that used to generate heat by thermogenesis (ucp1), and hunger suppressors (leptin) were all down-regulated (Fig. 2b, c). However, another lipase gene, bile salt-stimulated lipase (bssl), was upregulated in the tissue of the liver or intestine of HFD-fed bmp8a[-/-] zebrafish (Fig. 2d–g). The upregulation of the bssl1 and bssl2 in bmp8a[-/-] zebrafish is beneficial for lipid absorption, since Bssl is mainly involved in the hydrolysis of dietary fat. Hence, fat metabolism-related molecules are regulated by the bmp8a gene.

AMPK is known as the key mediator of fatty acid oxidation, so we further examined whether the activation of AMPK is involved in the regulatory process by Bmp8a. Notably, bmp8a knockout significantly decreased AMPK phosphorylation level in adipose and liver tissues (Fig. 2h–j). Meanwhile, the phosphorylation level of ACC was also reduced in bmp8a[-/-] zebrafish (Fig. 2h–j). It has known that the phosphorylation of ACC increases fatty acid oxidation by inhibiting the activity of ACC. Thus, bmp8a deletion can decrease fatty acid oxidation through reduced phosphorylation levels of AMPK and ACC. In addition, we found that the mRNA and protein levels of Pgc-1α and Ucp1 decrease in both adipose and liver tissues in bmp8a[-/-] zebrafish (Fig. 2b, c, h–j). AMPK has previously been reported to upregulate the abundance of PGC-1α, which can activate the transcription of UCP1 and other thermogenic genes[30]. Therefore, Bmp8a also regulates fatty acid oxidation through the AMPK-Pgc-1α-Ucp1 pathway.

We have proved that Bmp8a regulates immune responses through the p38 MAPK pathway[28]. Combined with other reports that the p38 MAPK-Pgc-1α-Ucp1 pathway plays a vital role in the oxidation of lipids[31], we hypothesized that Bmp8a might activate fatty acid oxidation through the p38 MAPK-Pgc-1α-Ucp1 pathway. Clearly, compared to wild-type ZFL cells, the phosphorylation level of p38 MAPK and the expression of Pgc-1α and Ucp1 proteins were increased in bmp8a-overexpressed ZFL cells (Fig. 2k–m).

Taken together, these results indicate that Bmp8a promotes fatty acid oxidation through AMPK and p38 MAPK-Pgc-1α-Ucp1 pathways. Our data provide a mechanistic explanation for how Bmp8a regulates fatty acid oxidation (Fig. 2n).

**Bmp8a inhibits adipocyte differentiation of 3T3-L1 cells.** There is no available information regarding the role of Bmp8a in adipocyte differentiation. Treating 3T3-L1 cells with methylisobutylxanthine, dexamethasone, and an insulin cocktail for two days (Day 0 ~ Day 2), followed by insulin treatment (Day 2 ~ Day 4),

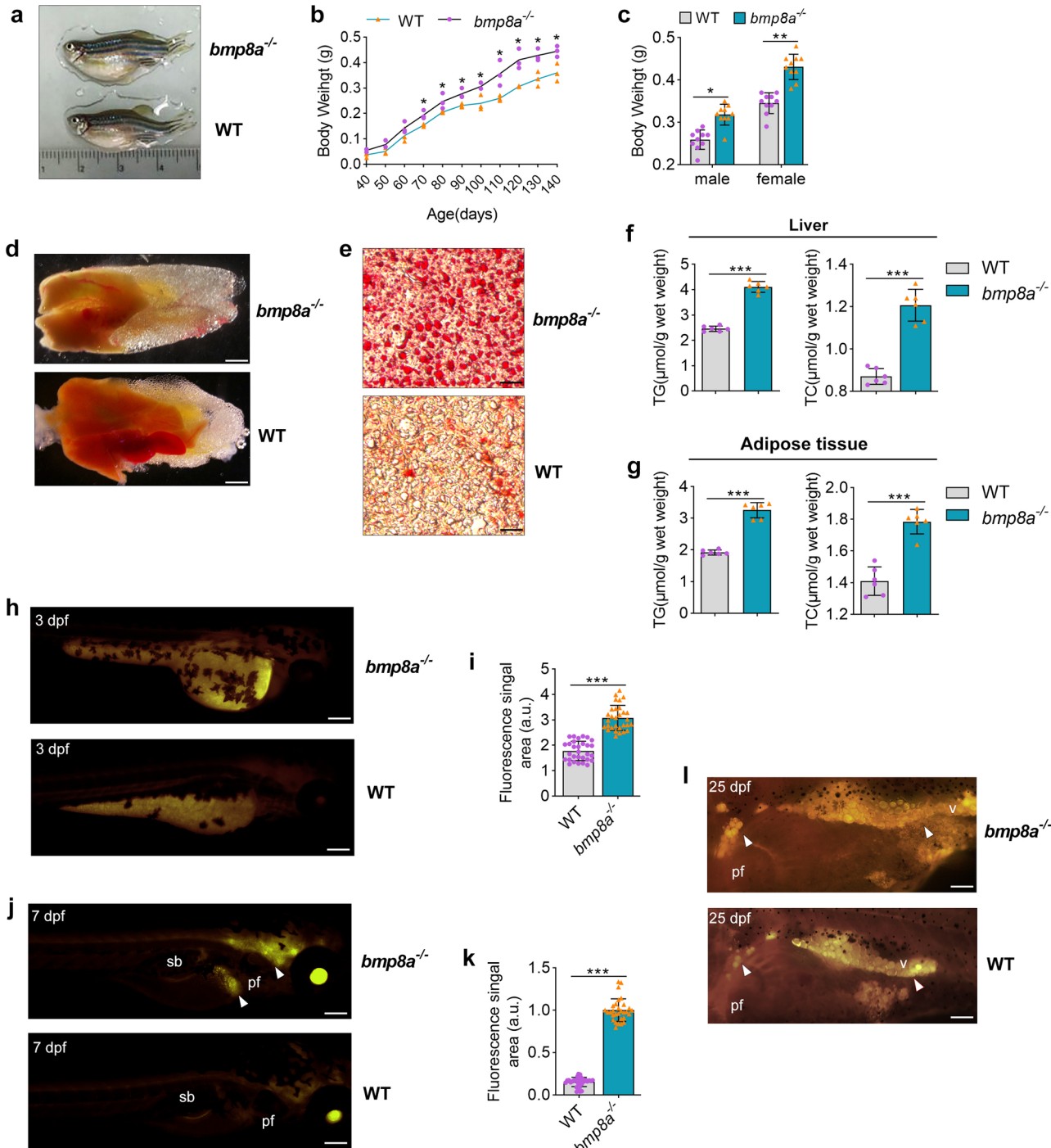

**Fig. 1 Obesogenic phenotype of _bmp8a⁻/⁻_ zebrafish. a** Representative photograph of WT and _bmp8a⁻/⁻_ zebrafish ($n = 10$). **b** Body weight changes of WT and _bmp8a⁻/⁻_ zebrafish ($n = 30$). **c** WT and _bmp8a⁻/⁻_ (male or female) zebrafish body weight at 140 days ($n = 10$). **d** Representative photographs of WT and _bmp8a⁻/⁻_ zebrafish visceral mass ($n = 6$). Scale bars = 1 mm. **e** Fatty liver changes were revealed by histopathological sections ($n = 6$). Scale bars = 20 μm. **f, g** The TG and TC level in WT and _bmp8a⁻/⁻_ zebrafish (**f**, $n = 6$) or adipose tissue (**g**, $n = 6$). **h–k** Representative photographs WT and _bmp8a⁻/⁻_ zebrafish at 3 dpf (**h**, **i**, $n = 30$) or 7dpf (**j**, **k**, $n = 30$) were stained with Nile Red. Scale bars = 300 μm. The fluorescence signal area was calculated by ImageJ software. **l** Representative Nile Red staining images of WT and _bmp8a⁻/⁻_ zebrafish at 25 dpf ($n = 10$). Scale bars = 300 μm. Swim bladder (sb), pectoral fin (pf), viscera (v). Data were representative of at least three independent experiments. Data were analyzed by two-tailed Student's _t_-test and presented as mean ± SD (*$p < 0.05$, **$p < 0.01$, ***$p < 0.001$).

these cells can be induced to differentiate into adipocyte-like cells. Intracellular lipid accumulation can be observed and quantified after staining with Oil Red O, providing an effective model system for adipogenesis in vitro (Fig. 3a). Firstly, we investigated the expression of _Bmp8a_ and adipogenic markers genes (_Pparγ and_

_C/ebpα_) during 3T3-L1 differentiation (Fig. 3b). Obviously, the expression of _Bmp8a_ was decreased at the later stage of adipocyte differentiation (Fig. 3b). Then, we examined the lipid accumulation using Oil-Red O staining to evaluate the effect of Bmp8a on 3T3-L1 adipocyte differentiation. Zebrafish _bmp8a_ or mouse

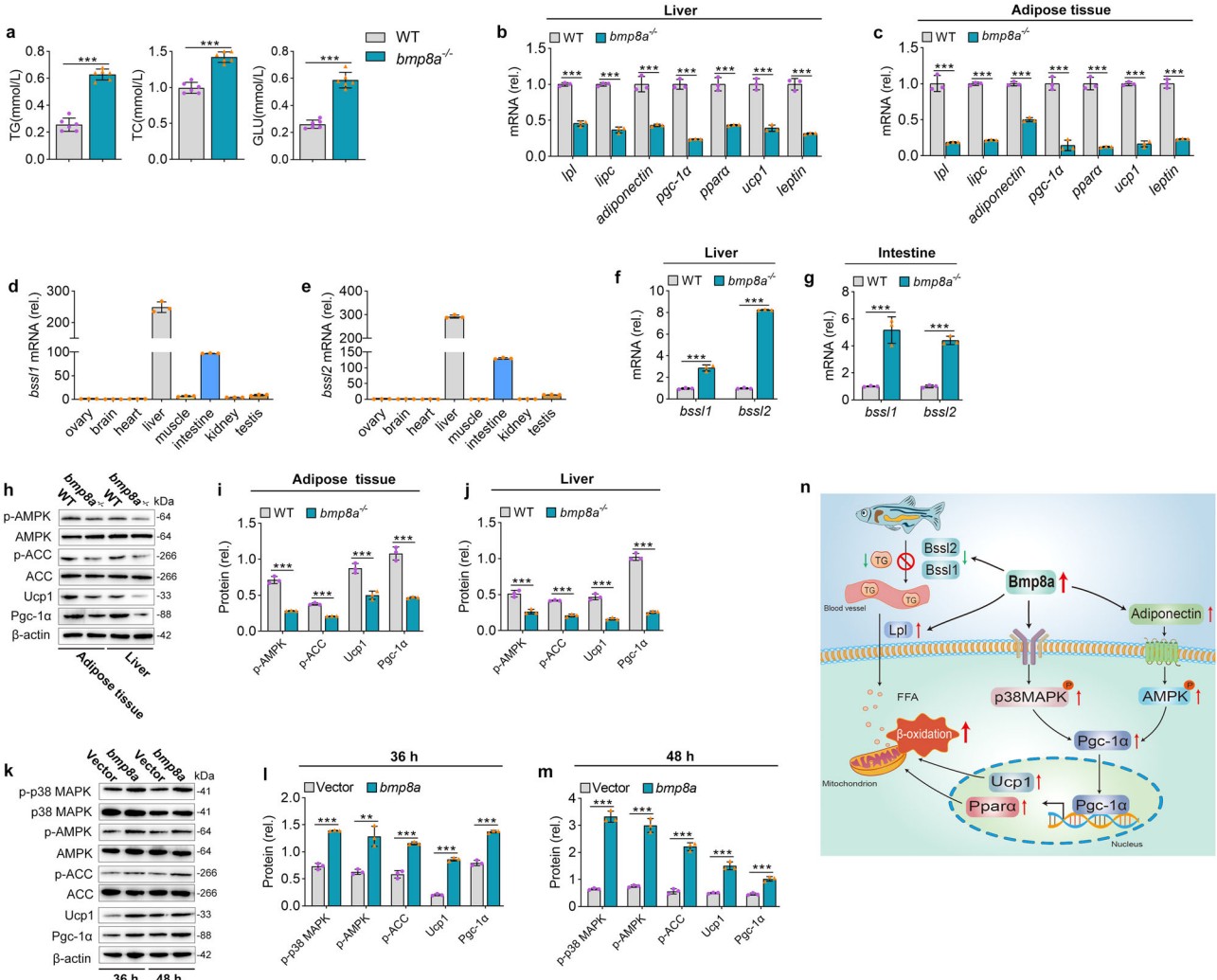

**Fig. 2 Bmp8a promotes fatty acid oxidation through AMPK and p38 MAPK pathways. a** The serum TG, TC and GLU level in WT and bmp8a$^{-/-}$ zebrafish ($n = 6$). **b, c** The qPCR analysis of genes related to fatty acid metabolism in the WT and bmp8a$^{-/-}$ zebrafish liver (**b**, $n = 3$) or adipose tissue (**c**, $n = 3$). **d, e** The bssl1 (**d**) and bssl2 (**e**) gene expression analysis in different zebrafish tissues ($n = 3$). **f, g** The qPCR analysis of bssl1 and bssl2 mRNA level in the liver (**f**, $n = 3$) and intestine (**g**, $n = 3$) from WT or bmp8a$^{-/-}$ zebrafish. **h–j** Validation and quantification of p-AMPK, p-ACC, Ucp1, and Pgc-1α expression in adipose tissue (**i**) and liver (**j**) from WT or bmp8a$^{-/-}$ zebrafish. Protein expression levels were quantified using ImageJ software and normalized to total protein or β-actin ($n = 3$). **k–m** Validation and quantification of p-p38 MAPK, p-AMPK, p-ACC, Ucp1, and Pgc-1α expression after overexpression of zebrafish bmp8a in ZFL cells. The cells were collected at 36 h (**l**) and 48 h (**m**) post-transfection for Immunoblot analysis. Protein expression levels were quantified using ImageJ software and normalized to total protein or β-actin ($n = 3$). **n** Schematic overview. Data were representative of at least three independent experiments. Data were analyzed by two-tailed Student's t-test and presented as mean ± SD (**$^{**}p < 0.01$, $^{***}p < 0.001$).

Bmp8a was successfully overexpressed or knocked down in 3T3-L1 cells (Fig. 3c, d). We found that overexpression of zebrafish bmp8a or mouse Bmp8a reduced lipid droplet formation (Fig. 3e, f). In contrast, lipid production increased when knocked down mouse Bmp8a (Fig. 4a, b). Also, overexpression of zebrafish bmp8a or mouse Bmp8a decreased the mRNA expression of adipogenic markers, such as C/ebpα, Pparγ, and Fasn (Fig. 3g–j). Protein levels of C/EBPα and PPARγ were also reduced in zebrafish bmp8a or mouse Bmp8a overexpressed 3T3-L1 cells (Fig. 3k–m). Furthermore, the knockdown of mouse Bmp8a caused a significantly increased mRNA expression of C/ebpα, Pparγ, and Fasn (Fig. 4c–f). Consistent with this, when Bmp8a was knocked down, PPARγ and C/EBPα protein levels increased (Fig. 4g–i). These results indicate that Bmp8a can inhibit adipocyte differentiation.

**Bmp8a activates Smad2/3 signaling to inhibit adipocyte differentiation in 3T3-L1 cells.** BMPs transmit signals through both Smad-dependent pathways (Smad1/5/8 and Smad2/3) and Smad-independent pathways (ERK, JNK, and p38 MAPK)[32–34]. We evaluated the activation of these signals by immunoblot assays in both Bmp8a-overexpressing and WT 3T3-L1 cells. Phosphorylation levels of Smad1/5/8 and Smad2/3, but not ERK, JNK, or p38 MAPK, were significantly increased in zebrafish bmp8a or mouse Bmp8a-overexpressing 3T3-L1 cells (Fig. 5a–d). Then, the effects of Smad1/5/8 inhibitor DMH1 and Smad2/3 inhibitor TP0427736 HCl on the lipid content were tested in Bmp8a-overexpressing 3T3-L1 cells. We found that only Smad2/3 inhibitor TP0427736 HCl blocked the decrease of lipid content in zebrafish bmp8a or mouse Bmp8a-overexpressing cells (Fig. 5e, f). Thus, it suggests that Bmp8a activates Smad2/3 signaling to inhibit preadipocyte differentiation in 3T3-L1 cells.

BMP binds to a heterotetrameric transmembrane receptor complex, which consists of two type I receptors (ALK2, ALK3, ALK4, ALK5, ALK6, and ALK7) and two type II receptors (BMPR2, ACVR2A, ACVR2B, and TGFβR2) (Fig. 5g)[35,36].

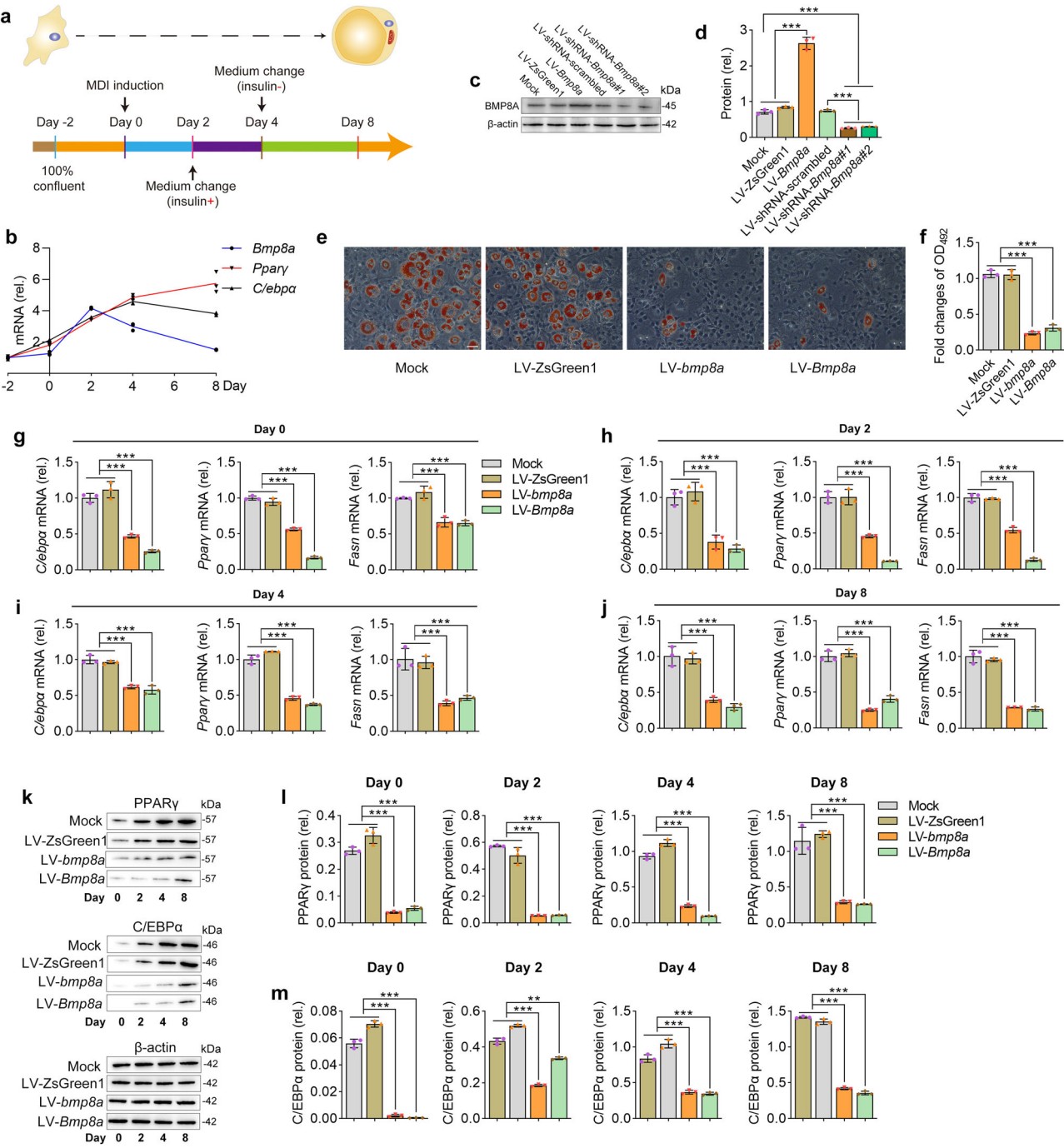

**Fig. 3 Stably overexpressing zebrafish *bmp8a* or mouse *Bmp8a* inhibits adipogenesis. a** Protocol for effective differentiation of 3T3-L1 cells into adipocytes. **b** The mRNA expression pattern of mouse *Bmp8a*, *Pparγ* and *C/ebpα* during 3T3-L1 cells differentiated into adipocytes (*n* = 3). **c, d** Immunoblot analysis of mouse BMP8A protein expression in 3T3-L1 cells (Mock), stably overexpressed empty plasmid in 3T3-L1 cells (LV-ZsGreen1), stably overexpressed mouse *Bmp8a* in 3T3-L1 cells (LV-*Bmp8a*), 3T3-L1 cells infected with scramble shRNA lentivirus (LV-shRNA-scrambled), and knockdown mouse *Bmp8a* in 3T3-L1 cells (shRNA-*Bmp8a#1* and shRNA-*Bmp8a#2*). BMP8A protein expression levels were quantified by ImageJ software and normalized to the amount of β-actin (**d**, *n* = 3). **e, f** After induction of adipogenic differentiation, differentiated 3T3-L1 adipocytes (Mock, LV-ZsGreen1, LV-*bmp8a*, and LV-*Bmp8a*) were stained with Oil Red O and subjected to OD$_{492}$ quantifications (*n* = 3). Scale bar = 20 μm. **g–j** On the day after induction as indicated, expressions of adipogenic genes (*Cebpα*, *Pparγ*, and *Fasn*) were examined at the mRNA level by qPCR (*n* = 3). **k–m** On the day after induction, as indicated, the protein levels of PPARγ and C/EBPα detected by Immunoblot. Protein expression levels were quantified using ImageJ software and normalized to the amount of β-actin (**l**, **m**, *n* = 3). Data were representative of at least three independent experiments. Data were analyzed by One-way ANOVA and presented as mean ± SD (**p < 0.01, ***p < 0.001).

Interestingly, the type I receptor *Alk6* gene was not found in the 3T3-L1 cells (Fig. 5h). Next, we quantified the relative abundance of transcripts encoding these receptors in 3T3-L1 cells, which *Alk3*, *Alk4*, and *Alk5* have the higher expression among the type I

receptor genes, while the expression of *Acvr2a*, *Bmpr2*, and *Tgfβr2* are higher among type II receptor genes (Fig. 5i). In parallel, we analyzed the expression patterns of these receptors during adipocyte differentiation (Supplementary Fig. 1a–i). The

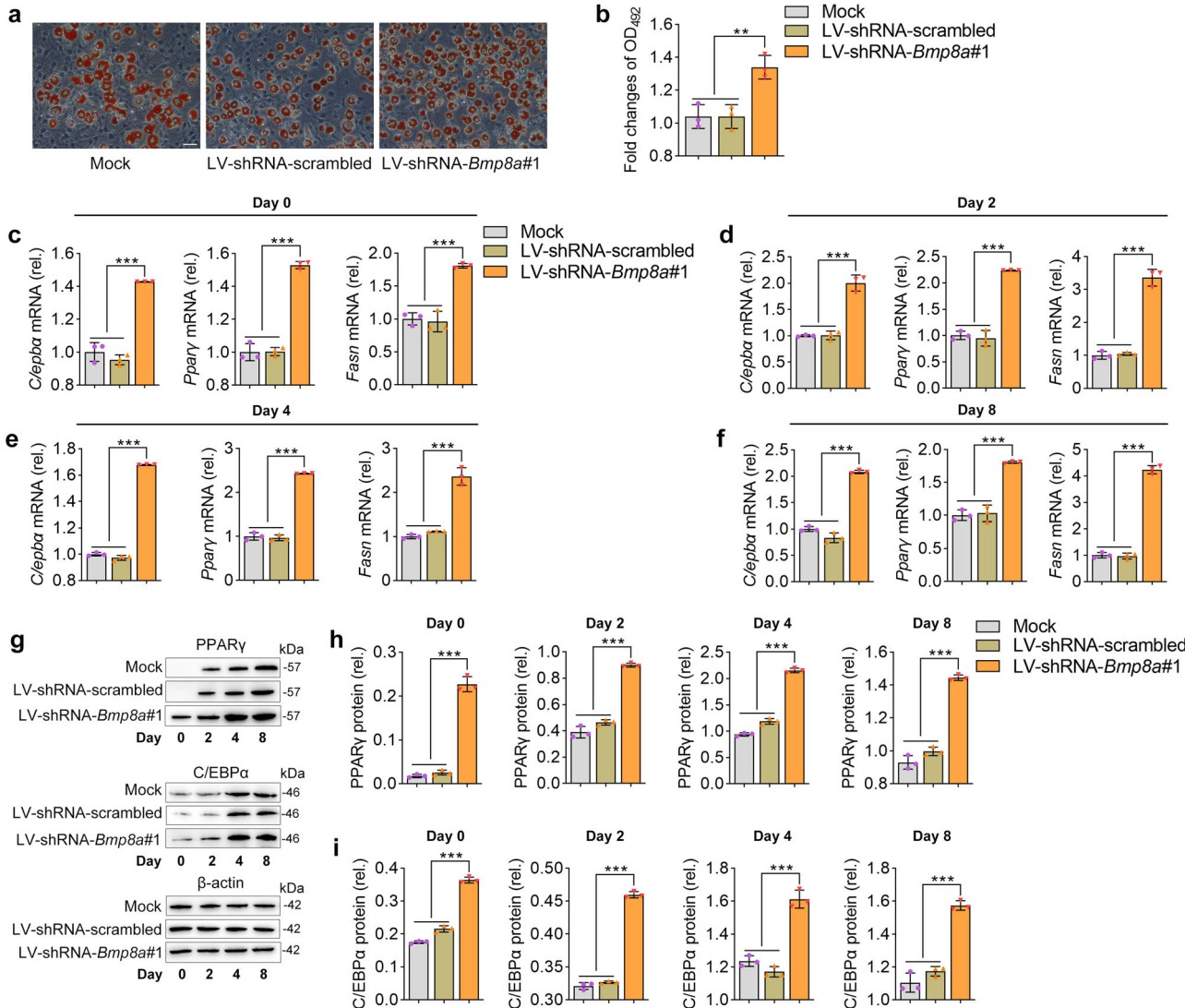

**Fig. 4 Knockdown *Bmp8a* promotes adipogenesis. a**, **b** After induction of adipogenic differentiation, *Bmp8a* knocked-down 3T3-L1 cells (LV-shRNA-*Bmp8a#1*) and control cells (Mock and LV-shRNA-scrambled) were stained with Oil Red O and subjected to OD$_{492}$ quantifications (n = 3). Scale bar = 20 μm. **c–f** On the day after induction as indicated, expressions of adipogenic markers (*Cebpα*, *Pparγ*, and *Fasn*) were examined at the mRNA levels by qPCR (n = 3). **g–i** On the day after induction as indicated, the protein levels of PPARγ and C/EBPα were detected by Immunoblot (**g**). Protein expression levels were quantified using ImageJ software and normalized to the amount of β-actin (**h**, **i**, n = 3). Data were representative of at least three independent experiments. Data were analyzed by One-way ANOVA and presented as mean ± SD (\*\*p < 0.01, \*\*\*p < 0.001).

expression pattern of type I receptors (*Alk2*, *Alk3*, *Alk4*, and *Alk5*) and type II receptors (*Acvr2a*, *Acvr2b*, *Bmpr2*, and *Tgfβr2*) during adipocyte differentiation were initially elevated and then progressively decreased. Also, the expression of the *Alk7* gene increased gradually alongwith adipocyte differentiation. To further examine the signal transduction pathway mediated by Bmp8a, BRE- and CAGA-driven luciferase reporter assays were performed. In this system, the BRE promoter was activated by signaling through Smad1/5/8, while the CAGA promoter was activated by signaling through Smad2/3[37]. Bmp8a could activate Smad1/5/8 signaling through receptor complexes formed by the type I receptor ALK3 and the type II receptor BMPR2 or ACVR2A (Fig. 5j, k). Also, Bmp8a was capable of activating Smad2/3 signaling through receptor complexes formed by the type I receptor ALK4 or ALK5 and the type II receptor ACVR2A, ACVR2B, or TGFβR2 (Fig. 5l, m). Notably, the CAGA-driven luciferase reporter system exhibited higher potency than the BRE-driven luciferase reporter system, indicating that the activation of the Smad2/3 signal by Bmp8a plays a dominant role. These

studies further support the conclusion that Bmp8a inhibits adipocyte differentiation through Smad2/3 signaling in 3T3-L1 cells.

**Bmp8a activates Smad2/3 signaling to inhibit adipogenesis through type I receptor ALK4.** As described above (Fig. 5j–m), it appears that the type I receptors ALK3, ALK4, and ALK5 are activated by Bmp8a. Here we further explored the role of these receptors in Bmp8a-mediated inhibition of adipocyte differentiation. Given the Gly-Ser (GS) domain of type I receptor requires phosphorylation for activation, we constructed dominant-negative mutants (*Alk3-ΔGS*, *Alk4-ΔGS*, and *Alk5-ΔGS*) plasmid (Fig. 6a). Lentiviral vectors *Alk3-ΔGS*, *Alk4-ΔGS*, or *Alk5-ΔGS* were transfected into LV-*bmp8a* and LV-*Bmp8a* 3T3-L1 cells, respectively. Knockdown of ALK3, ALK4, or ALK5 reduced Bmp8a-mediated phosphorylation of the Smad1/5/8 or Smad2/3 (Fig. 6b–g). However, the lipid accumulation test using Oil-Red O staining displayed that only the knockdown *of Alk4*

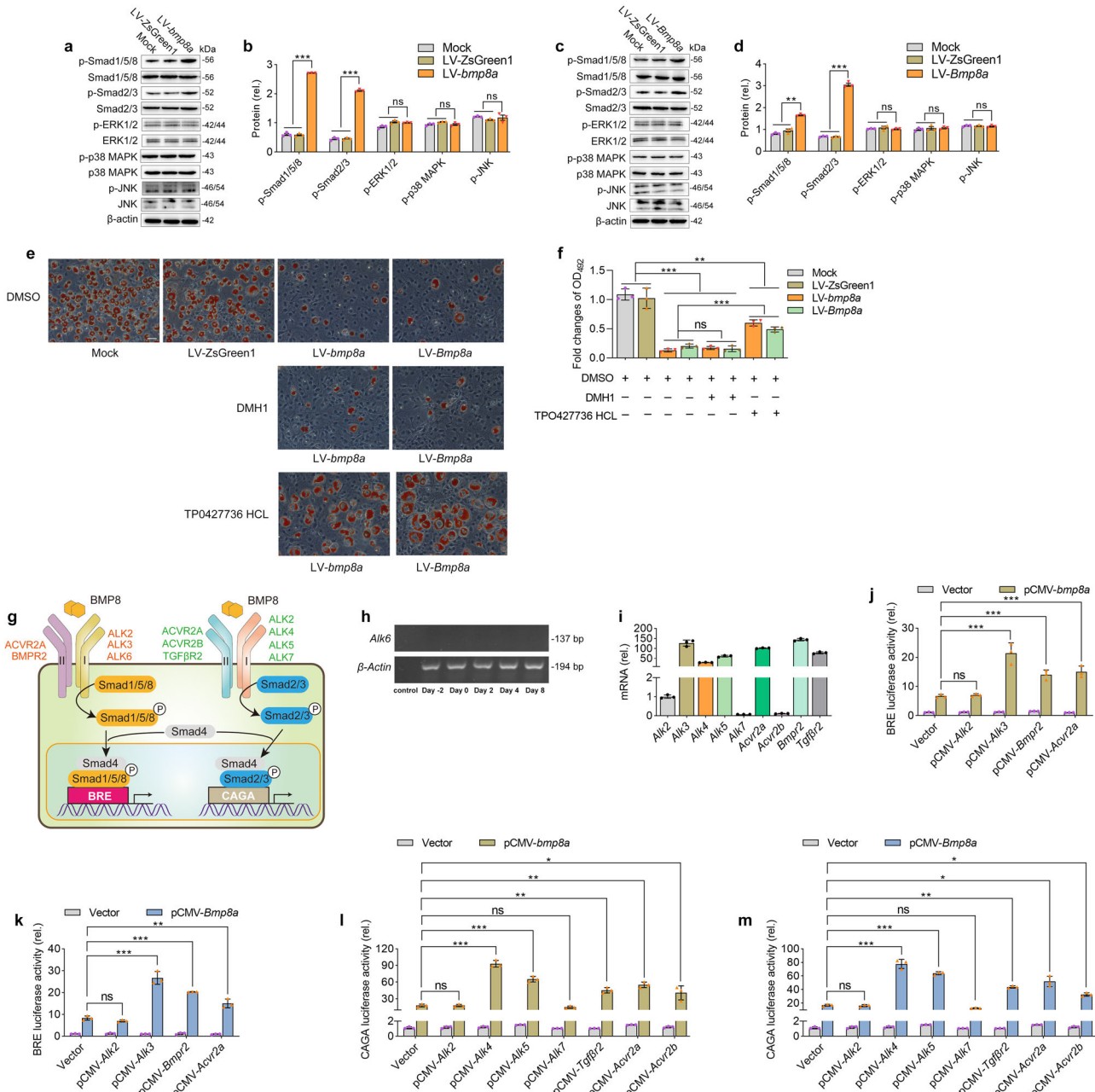

**Fig. 5 Bmp8a activates Smad2/3 signaling to inhibit adipocyte differentiation in 3T3-L1 cells. a–d** Representative western blot analysis and quantification of changes in p-Smad1/5/8, p-Smad2/3, p-ERK1/2, p-p38 MAPK, and p-JNK expression in LV-bmp8a cells (**a**, **b**) or LV-Bmp8a cells (**c**, **d**). Protein expression levels were quantified using ImageJ software and normalized to the amount of total protein ($n = 3$). **e, f** Representative Oil Red O staining photographs of LV-bmp8a and LV-Bmp8a 3T3-L1 cells were induced to adipogenic in the presence of DMH1 or TP0427736 HCL, dimethylsulfoxide (DMSO) as a vehicle and subjected to $OD_{492}$ quantifications ($n = 3$). Scale bar = 20 μm. **g** Schematic diagram of BMP8 mediated signal transduction. BMP8 can activate Smad1/5/8 signal transduction through the receptor complex formed by type I receptor ALK2, ALK3, or ALK6 and type II receptor ACVR2A or BMPR2. Meanwhile, BMP8 can also activate Smad2/3 signal transduction through the receptor complex formed by type I receptors ALK4 or ALK5 and type II receptors ACVR2A, ACVR2B, or TGFBR2. **h** Non-expression of mouse Alk6 gene in 3T3-L1 cells ($n = 3$). **i** The qPCR quantification of the type I receptor (Alk2, Alk3, Alk4, Alk5, Alk7) and type II receptor (Acvr2a, Acvr2b, Bmpr2, Tgrβr2) transcripts expressed in 3T3-L1 cells ($n = 3$). **j, k** Quantification of the activity of BRE-driven luciferase reporters with pCMV-bmp8a (**j**) or pCMV-Bmp8a (**k**) cotransfected with pCMV-Alk2, pCMV-Alk3, pCMV-Bmpr2, pCMV-Acrv2a, respectively ($n = 3$). Renilla luciferase was used as the internal control. **l, m** Quantification of the activity of CAGA-driven luciferase reporters with pCMV-bmp8a (**l**) or pCMV-Bmp8a (**m**) cotransfected with pCMV-Alk2, pCMV-Alk3, pCMV-Bmpr2, and pCMV-Acrv2a, respectively ($n = 3$). Renilla luciferase was used as the internal control. Data were representative of at least three independent experiments. Data were analyzed by One-way ANOVA and presented as mean ± SD (ns not significant, *$p < 0.05$, **$p < 0.01$, ***$p < 0.001$).

significantly alleviated the ability of Bmp8a to inhibit the differentiation of preadipocytes into mature adipocytes (Fig. 6h, i). The data suggest that ALK4 is involved in the Bmp8a-Smad2/3 inhibition pathway of adipogenesis.

**Smad2/3 binds to Pparγ promoter to inhibit its transcription.** Although many studies aim to understand the role of Smad2/3 signaling in inhibiting adipogenesis, it remains unclear whether Smad2/3 binds to the Pparγ promoter to repress its transcription.

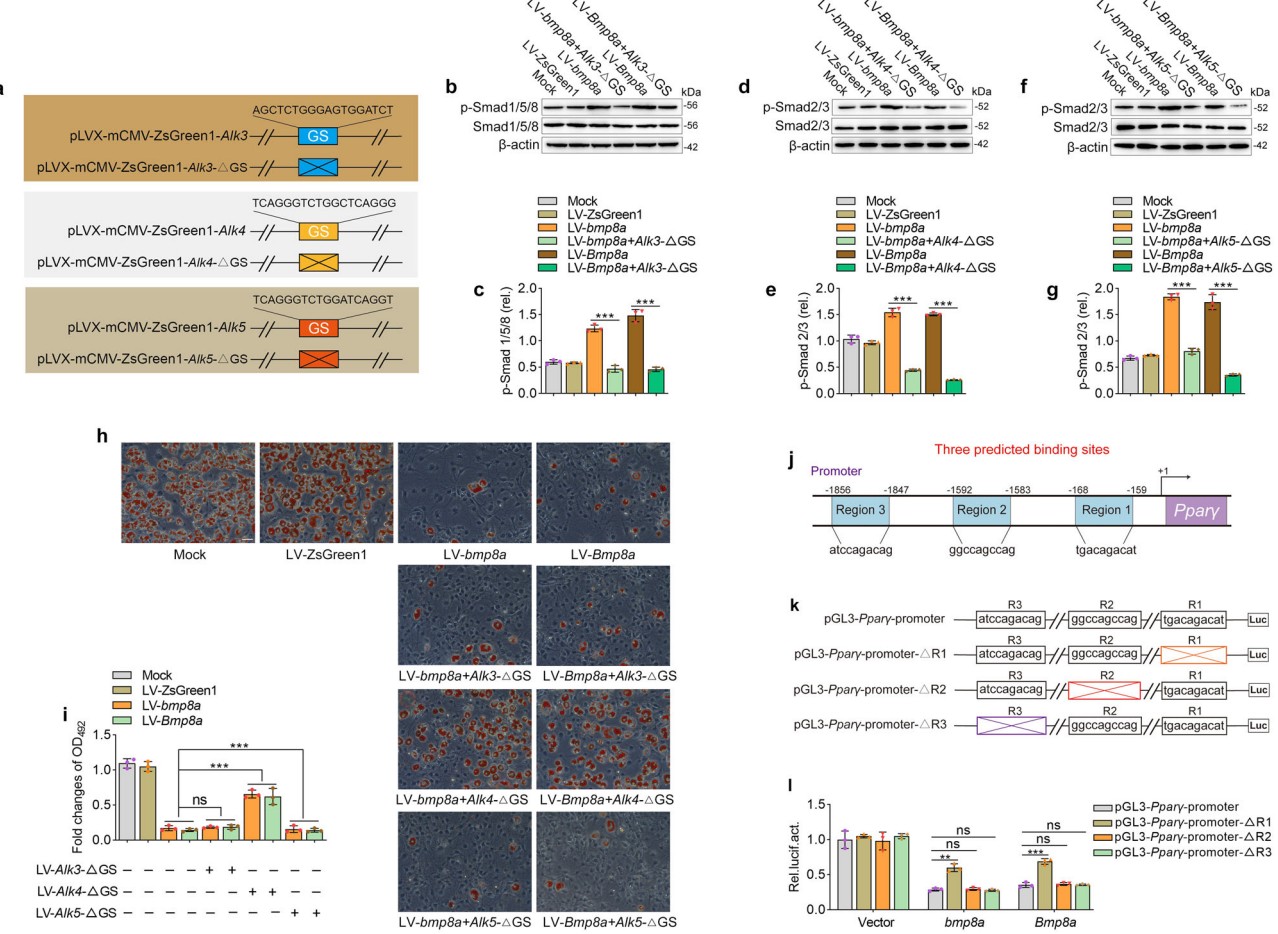

**Fig. 6 Bmp8a activates Smad2/3 signaling to inhibit adipogenesis through type I receptor ALK4. a** Schematic drawing of wild-type and GS motif mutation (*Alk3*-ΔGS, *Alk4*-ΔGS, *Alk5*-ΔGS) plasmids. **b**, **c** Immunoblot analysis and quantification of p-Smad1/5/8 in Mock, LV-ZsGreen1, LV-*bmp8a*, LV-*bmp8a* + *Alk3*-ΔGS, LV-*Bmp8a*, and LV-*Bmp8a* + *Alk3*-ΔGS 3T3-L1 cells (*n* = 3). **d**, **e** Immunoblot analysis and quantification of p-Smad1/5/8 in Mock, LV-ZsGreen1, LV-*bmp8a*, LV-*bmp8a* + *Alk4*-ΔGS, LV-*Bmp8a*, and LV-*Bmp8a* + *Alk4*-ΔGS 3T3-L1 cells (*n* = 3). **f**, **g** Immunoblot analysis of p-Smad1/5/8 in Mock, LV-ZsGreen1, LV-*bmp8a*, LV-*bmp8a* + *Alk5*-ΔGS, LV-*Bmp8a*, and LV-*Bmp8a* + *Alk5*-ΔGS 3T3-L1 cells (*n* = 3). Protein expression levels were quantified by using ImageJ software and normalized to the amount of total protein. **h**, **i** knocked-down ALK3, ALK4, and ALK5 in LV-*bmp8a* or LV-*Bmp8a* 3T3-L1 cells, were induced to differentiate. Lipid contents of the resulting adipocyte-like cells were stained and quantified (*n* = 3). **j** Schematic diagram of the *Pparγ* promoter region. Three predicted TF binding sites and sequences. **k** Schematic drawing of wild-type and predicted TF binding sites mutation plasmids (pGL3-*Pparγ*-promoter-ΔR1, pGL3-*Pparγ*-promoter-ΔR2, pGL3-*Pparγ*-promoter-ΔR3). **l** Dual-luciferase report assay was used to analyze the abilities of zebrafish *bmp8a* and mouse *Bmp8a* in activation of the *Pparγ* promoter (*n* = 3). The pGL3-*Pparγ*-promoter, pGL3-*Pparγ* -promoter-ΔR1, pGL3-*Pparγ*-promoter-ΔR2 or pGL3-*Pparγ*-promoter-ΔR3 was transfected into HEK293T cells along with pCMV-*bmp8a*, pCMV- *Bmp8a* or empty vector. After 48 h, the transfected cells were collected for luciferase assays. *Renilla* luciferase was used as the internal control. Data were from three independent experiments and were analyzed by One-way ANOVA and were presented as mean ± SD (ns not significant, **$p < 0.01$, ***$p < 0.001$).

To address this question, we used an online promoter prediction tool (http://jaspar.genereg.net/) to search for potential Smad2/3 transcription factor (TF) binding sites in the *Pparγ* promoter. There are three potential TF binding sites with high scores according to the prediction (Fig. 6j). We constructed a series of luciferase plasmids containing the pGL3-*Pparγ*-promoter and three TF binding site mutant plasmids (pGL3-*Pparγ*-promoter-ΔR1, pGL3-*Pparγ*-promoter-ΔR2, and pGL3-*Pparγ*-promoter-ΔR3) (Fig. 6k). We found that Smad2/3 did repress *Pparγ* transcription and deletion of the Region 1 sequence (pGL3-*Pparγ*-promoter-ΔR1) in the *Pparγ* promoter attenuated the ability of Bmp8a to suppress *Pparγ* promoter activity, suggesting that Region 1 may be the region where Smad2/3 binds to the *Pparγ* promoter (Fig. 6l).

**An implication of functional bridge between immune regulation and adipocyte differentiation**. To further explore the

molecular mechanism by which Bmp8a inhibits adipogenesis, we performed transcriptome analysis of the LV-*bmp8a*, LV-*Bmp8a*, and LV-ZsGreen1 3T3-L1 cells. To identify the differentially expressed genes (DEGs) in the two cell types (LV-*bmp8a* or LV-*Bmp8a*), the cutoff values for the fold change and *P* value were set to 2.0 and 0.05, respectively. Among the DEGs, 2337 genes (1215 downregulated genes and 1122 upregulated genes) were significantly modulated in LV-*bmp8a* cells, while 2187 genes (1211 downregulated genes and 976 upregulated genes) were modulated in LV-*Bmp8a* cells. By combining the two data sets, we found the two cell types shared 536 overlapping downregulated DEGs and 334 overlapping upregulated DEGs (Supplementary Fig. 2 and Supplementary Fig. 3). These results suggest that overexpression of zebrafish *bmp8a* or mouse *Bmp8a* in 3T3-L1 cell lines exhibits a comparable transcriptional response.

Kyoto Encyclopedia of Genes and Genomes (KEGG) analyses revealed that compared with LV-ZsGreen1 3T3-L1 cells, the

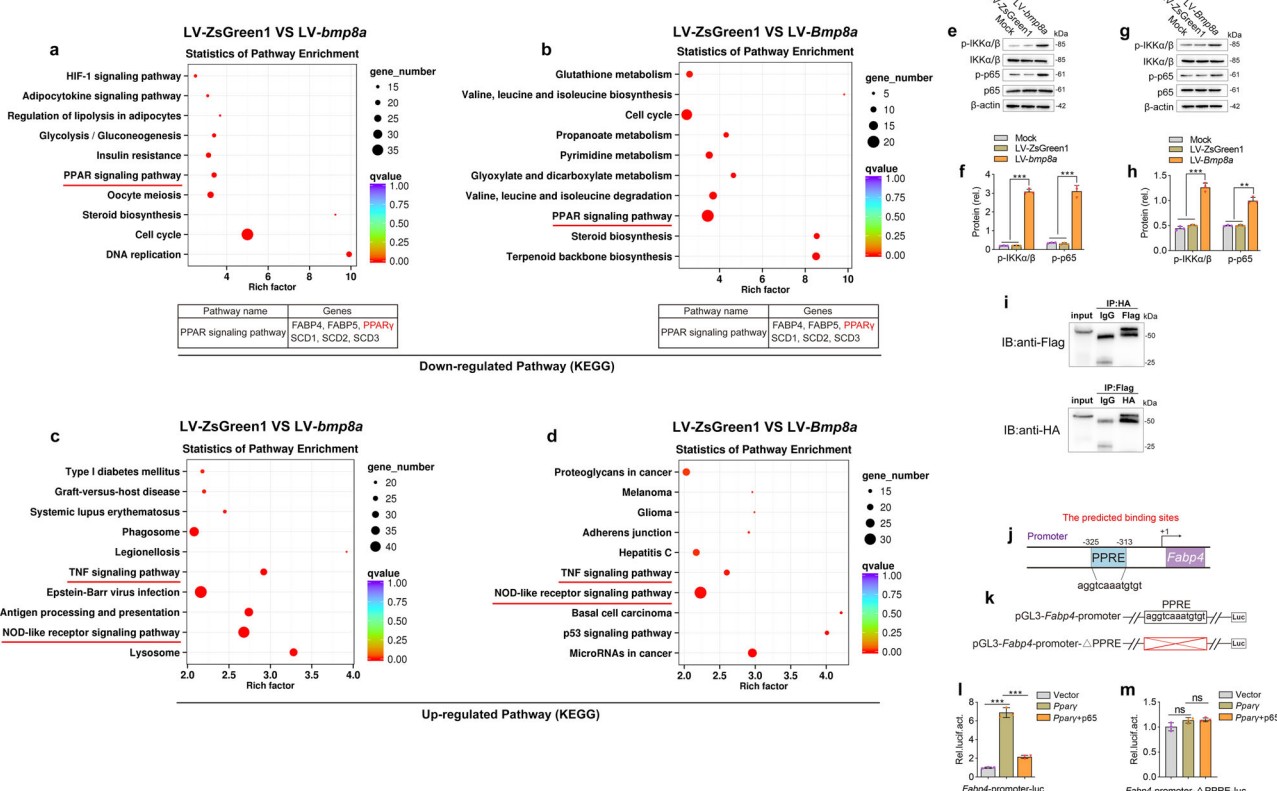

**Fig. 7 The interaction of NF-κB and PPARγ mediates the effect of Bmp8a on adipogenesis. a, c** After induction of adipogenic differentiation, the downregulated (**a**) and upregulated (**c**) KEGG pathway in overexpression zebrafish *bmp8a* 3T3-L1 cells. **b, d** After induction of adipogenic differentiation, the downregulated (**b**) and upregulated (**d**) KEGG pathway in overexpression mouse *Bmp8a* 3T3-L1 cells. **e, f** Immunoblot analysis and quantification of p-IKKα/β and p-p65 in Mock, LV-ZsGreen1, and LV-*bmp8a* 3T3-L1 cells (*n* = 3). **g, h** Immunoblot analysis and quantification of p-IKKα/β and p-p65 in Mock, LV-ZsGreen1, and LV-*Bmp8a* 3T3-L1cells. Protein expression levels were quantified by ImageJ software and normalized to total protein (*n* = 3). **i** Co-immunoprecipitation and immunoblot analysis of co-transfected with PPARγ and p65 (*n* = 3). **j** Schematic drawing of predicted PPRE site in *Fabp4* promoter region. **k** Schematic drawing of WT and PPRE site mutation Luc-report plasmids. **l, m** Quantification of the activity of *Fabp4*-promoter (**l**) and *Fabp4*-promoter-ΔPPRE (**m**) luciferase reporters in mouse HEK293T cells transfected with Vector, pCMV-*Pparγ*, or co-transfected pCMV-*Pparγ* and pCMV-p65, respectively. *Renilla* luciferase was used as the internal control (*n* = 3). Data were from three independent experiments and were analyzed by One-way ANOVA and were presented as mean ± SD (ns not significant, **$p < 0.01$, ***$p < 0.001$).

downregulated genes were remarkably enriched in the PPAR signaling pathway whether in LV-*bmp8a* and LV-*Bmp8a* 3T3-L1 cells (Fig. 7a, b). More intriguingly, KEGG analyses showed that the upregulated genes remarkably enriched in NOD-like receptor signaling and TNF signaling pathways involved in the immune process in LV-*bmp8a* or LV-*Bmp8a* 3T3-L1 cells, compared with LV-ZsGreen1 3T3-L1 cells (Fig. 7c, d). Given that NF-κB molecules are downstream of NOD-like receptor signaling and TNF signaling pathways, we wondered whether Bmp8a could activate the NF-κB signaling. It was confirmed that phosphorylation levels of IKKα/β and p65 increased in LV-*bmp8a* or LV-*Bmp8a* 3T3-L1 cells (Fig. 7e–h).

We have shown that Bmp8a inhibits the expression of *Pparγ*, so we want to understand the relationship between NF-κB and PPARγ in regulating adipogenesis. Previous studies have revealed that NF-κB components p50 and p65 bind PPARγ directly in vitro by GST Pull-down assay[38]. Our co-immunoprecipitation (Co-IP) experiments confirmed that NF-κB (p65) interacts with PPARγ (Fig. 7i). We suspect that the interaction between NF-κB and PPARγ blocks PPARγ activation of its target genes, leading to inhibition of adipogenesis. To our knowledge, PPARγ regulates the expression of target genes by binding to peroxisome proliferator response elements (PPRE) in their promoters[39]. Fatty acid binding protein 4 (FABP4), as a target gene of PPARγ, promotes adipocyte differentiation[40]. We found that a putative

PPRE is present in the promoter region of *Fabp4* (Fig. 7j). Not surprisingly, overexpression of PPARγ stimulated FABP4 expression. However, overexpression of both PPARγ and p65 impaired the ability of PPARγ to activate FABP4 expression (Fig. 7l). Deletion of PPRE in *Fabp4* promoter prevented PPARγ from activating FABP4 expression (Fig. 7m). Overall, these results indicate that Bmp8a regulates PPARγ activity through NF-κB signaling to inhibit adipocyte differentiation.

## Discussion
The BMPs are members of a large highly conserved family of extracellular polypeptide signaling molecules of the TGF-β superfamily. In *Mus musculus*, there are two members of the *Bmp8* gene, *Bmp8a* and *Bmp8b*, which arose from a recent duplication of a single gene[41]. However, only a single *bmp8a* gene is present in *D. rerio*[27]. BMP8B is induced by nutritional and thermogenic factors in mature brown adipose tissue (BAT), increasing the response to noradrenaline through enhanced p38MAPK/CREB signaling and increased lipase activity. *Bmp8b*⁻/⁻ mice exhibit impaired thermogenesis and reduced metabolic rate, causing weight gain[23]. BMP8A is almost absent from BAT, but enriched in white adipose tissue (WAT)[23]. There have been no reports on the role of BMP8A in adipose tissue. In this study, we showed that zebrafish Bmp8a or mouse BMP8A has an anti-fat effect by promoting fatty acid oxidation and reducing adipocyte differentiation.

Zebrafish as a model organism in many fields of research, is becoming an increasingly powerful tool in lipid research since the lipid metabolic pathway between fish and mammals is conserved[29,42–44]. We have previously shown that *bmp8a* mRNA expression in the intestine or brain significantly upregulated in obese zebrafish induced by a high fat diet[27]. In this study, we found that compared to WT, *bmp8a*[−/−] zebrafish exhibited higher body weight and increased fat production, confirming the linking of Bmp8a with obesity. Furthermore, we showed that the expression of key genes associated with lipid metabolism (*lpl, lipc, adiponectin, pgc-1α*, and *pparα*), thermogenesis (*ucp1*), and appetite (*leptin*) were regulated by Bmp8a. Thus, Bmp8a appears to regulate obesity through multiple molecular pathways. Overall, we speculate that Bmp8a increases lipase activities such as Lpl and Lipc to hydrolyze triglyceride (TG) into free fatty acids (FFA). Subsequently, Bmp8a enhances FFA oxidation through AMPK or p38 MAPK pathway, thereby reducing lipid accumulation (Fig. 2n).

We also found fatty liver in *bmp8a*[−/−] zebrafish. The liver is a crucial player in regulating lipid metabolism throughout the body. The lipogenesis, adipolysis, lipoprotein synthesis, and secretion are mainly carried out in the liver[45]. Meanwhile, dysregulation of lipid metabolism is increasingly recognized as a hallmark of obesity and non-alcoholic fatty liver disease (NAFLD)[46,47]. However, we are unsure about the causal relationship between the fatty liver and dysregulation of lipid metabolism in *bmp8a*[−/−] zebrafish. Most likely, these two aspects lead and influence each other.

The role of BMP8A in regulating adipocyte differentiation is not yet clear. A series of transcriptional events coordinate the differentiation from preadipocytes to mature adipocytes[48]. Adipogenic factors C/EBPα and PPARγ are two central players in white adipocyte differentiation[49]. Here, we illustrated the inhibitory regulation of Bmp8a on adipogenesis by decreasing the expression of adipogenic markers (C/EBPα and PPARγ). Also, the expression of *pparγ* was increased in both liver and adipose tissue of *bmp8a*[−/−] zebrafish compared to wild-type zebrafish (Supplementary Fig. 4). Therefore, it is conceivable that obesity could result if the inhibitory regulation of adipogenesis by Bmp8a is disrupted, which is consistent with our findings of significantly increased body weight in *bmp8a*[−/−] zebrafish. Here the upregulation of the expression of *pparγ* does not seem to reconcile with the reduced expression of *adiponectin* and *lpl* in obese mutant zebrafish, but there are other comparable reports. For example, it was found that the adiponectin level in obese patients was significantly lower than in non-obese people[50,51]. Also, obesity can increase Bmp8a expression[27].

BMP achieves its signaling activity by interacting with the heterotetrameric receptor complex of transmembrane serine/threonine kinase receptors, BMPR type I and BMPR type II. ALK2, ALK3, ALK4, ALK5, ALK6 and ALK7 were identified as BMPR type I, while BMPR2, ACVR2A, ACVR2B, TGFβR2 were identified as BMPR type II[35,36]. The *Alk6* gene was not found in 3T3-L1 cells, although ALK6 involved in the antivirus immunity in zebrafish[28]. Therefore, different BMP functions are achieved by binding to different BMP receptors. The activated receptor kinases are well known to transmit signals through Smad-dependent pathways, including Smad1/5/8 and Smad2/3 pathways, as well as Smad-independent pathways, including ERK, JNK, and p38 MAPK pathways[32–34]. It has been reported that Smad1/5/8 signaling is fundamental for priming and driving the commitment of 3T3-L1 cells toward adipogenic fates, whereas Smad2/3 activation may blunt adipogenesis via a negative feedback loop that reduces Smad1/5/8 signaling[52]. In this study, we found that Bmp8a inhibits adipogenesis by activating Smad2/3 signaling. We further showed that Smad2/3 could directly bind

to the *Pparγ* promoter to inhibit its transcription. To our puzzled, Bmp8a could increase the phosphorylation level of Smad1/5/8, but Smad1/5/8 inhibitor DMH1 has no significant impact on the decrease of lipid content in *Bmp8a*-overexpressing 3T3-L1 cells. It would be interesting to gain a comprehensive understanding of the mechanism of regulation of Bmp8a on preadipocyte differentiation.

Recently, studies have found that BMP6 activity in the liver has a positive immune function[53]. Also, the NK cell-mediated cytotoxic signaling pathway in the liver of *Bmp9*[−/−] mice was affected[26]. Here we found that Bmp8a could increase NOD-like receptor signaling and TNF signaling in 3T3-L1 cells, indicating the role of Bmp8a in regulating the immune process. Adipocytes have an innate antiviral system that regulates adipocyte function[54]. The preadipocytes express antiviral pattern recognition receptors such as TLR3, MDA5, and RIG-I, which can respond to the virus by producing IL6, TNFa and type I IFNs[54]. Meanwhile, virus stimulation also inhibits the differentiation of preadipocytes into adipocytes[54]. This finding can be explained from an evolutionarily view of the host response; that is, to inhibit adipocyte differentiation and conserve energy against infection under virus stimulation[55]. However, the mechanism by which virus stimulation inhibits preadipocyte differentiation remains unclear. NF-κB is a downstream molecule of both NOD-like receptor signaling and TNF signaling pathways. We further revealed that Bmp8a could activate the NF-κB signaling. It has been reported that NF-κB inhibits the expression of PPARγ[56], but the mechanism by which it inhibits PPARγ expression remains unclear. We confirmed the interaction of NF-κB and PPARγ, and the binding blocked PPARγ to activate its target gene *FABP4*, thereby inhibiting adipocyte differentiation. The interaction of NF-κB and PPARγ provides a functional linker between immune regulation and adipocyte differentiation (Fig. 8). In addition, it should also note that since NF-κB is an inflammatory signaling, it would be interesting to check what is the inflammatory status of adipose tissue and liver of mutant *bmp8a*[−/−] zebrafish. We found that the expression of pro-inflammatory genes, *tnfα* and *il-1β*, were elevated in *bmp8a*[−/−] zebrafish liver and adipose tissue (Supplementary Fig. 5). Considering the pro-inflammatory cytokines are generally produced in obesity[57], our data are in line with our observation that a prominent fatty liver appeared in *bmp8a*[−/−] zebrafish (Fig. 1e). Thus, we provided the evidence on the interesting effect of Bmp8a not only on adipogenesis, but also on inflammation.

In conclusion, we reported that *bmp8a*[−/−] zebrafish displayed obesity and fatty liver by decreasing fatty acid oxidation via downregulation of the phosphorylation of AMPK and ACC. Bmp8a suppresses the differentiation of 3T3-L1 preadipocytes into mature adipocytes by increasing the Smad2/3 signal. Bmp8a overexpression markedly increases NOD-like receptor signaling and TNF signaling pathways in 3T3-L1 cells. Furthermore, NF-κB interacts with PPARγ, providing a signaling bridge between immune regulation and adipocyte differentiation. We bring a previously unidentified insight into Bmp8a-mediated adipogenesis. These findings will provide a window into adipose development and metabolism, and bring a basis for the development of strategies targeting obesity and metabolic imbalance.

## Methods

**Zebrafish**. All animal experiments were performed in accordance with the Institutional Animal Care and Use Committee of the Ocean University of China (SD2007695). The zebrafish *bmp8a* gene homozygous mutant lines (*bmp8a*[−/−]) were established from the zebrafish AB line using TALENs technology[28]. Embryos from natural matings were grown at 28 °C. Female zebrafish were used for the experiments unless otherwise specified.

**Cell culture**. Mouse preadipocytes 3T3-L1 and human HEK293T cells were obtained from ATCC. The zebrafish liver cells (ZFL) were acquired from the China

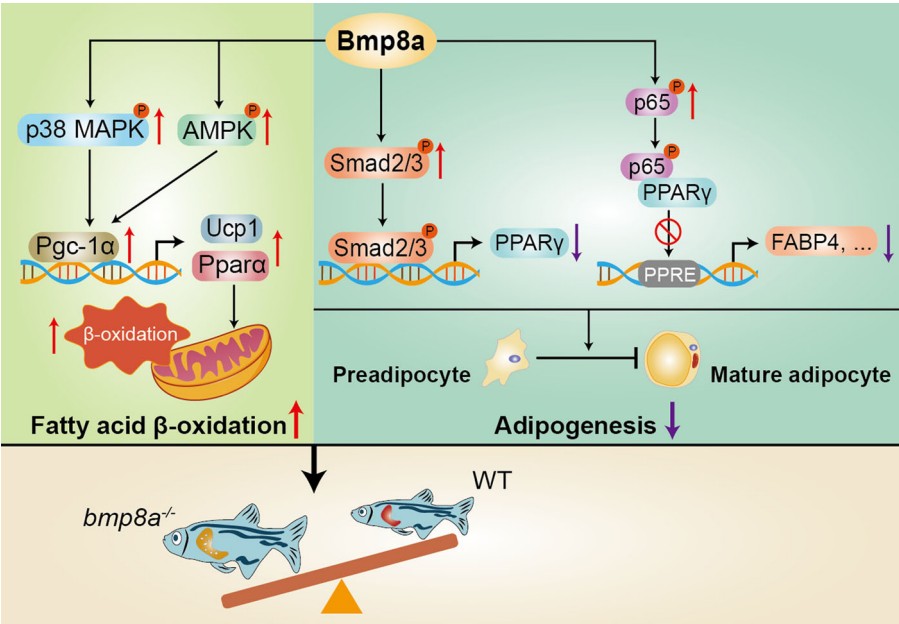

**Fig. 8 Schematic illustration of Bmp8a effect on the regulation of lipid metabolism and adipocyte differentiation.** Bmp8a is required for fatty acid oxidation and regulates adipogenesis as evidenced by weight gain and fatty liver of zebrafish lacking Bmp8a.

Zebrafish Resource Center (CZRC). 3T3-L1 and HEK293T cells were maintained in Dulbecco's modified Eagle's medium (DMEM, VivaCell, #C3113-0500) supplemented with 10% fetal bovine serum (FBS, Gibco, #26140-097) and penicillin/streptomycin at 37 °C in a humidified incubator with 5% $CO_2$. The ZFL cells were maintained in DMEM/F-12 media supplemented with 10% FBS and penicillin/streptomycin at 37 °C in a humidified incubator with 5% $CO_2$. All cell lines were subjected to the examination of mycoplasma contamination and were cultured for no more than 1 month. The cell morphology was confirmed periodically to avoid cross-contamination or misuse of cell lines.

**Induction of differentiation in 3T3-L1 cells.** Briefly, two days after reaching 100% confluency (Day 0), 3T3-L1 preadipocytes were treated with MDI cocktail, including 1 µg/ml insulin (beyotime, #P3376), 0.25 µM dexamethasone (DEX, Sigma-Aldrich, #D4902), and 0.5 mM isobutylmethylxanthine (IBMX, Sigma-Aldrich, #SI-I7018) for 48 h. Then, cells were transferred to DMEM with insulin (1 µg/ml) and 10% FBS for 2 days. Afterward, cells were cultured in DMEM containing 10% FBS and the medium was replaced every 2 days until more than 90% of cells showed adipocyte morphology.

**Lentiviral production and stable transfection cells.** HEK-293T cells were co-transfected with Lentiviral vectors (pLVX-mCMV-ZsGreen1-Puro, Youbio, #VT8138; pLVX-shRNA2, Youbio, #VT1457) and packaging vectors (pMD2.G, Addgene, #12259; psPAX2, Addgene, #12260) using FuGENE HD Transfection Reagent (Promega, #E2311) were packaged in HEK293T cells and concentrated using the Lentivirus Concentration Solution Kit (Yeasen, #41101ES50). After transfection into 3T3-L1 cells, 2 µg/ml puromycin (Solarbio, #P8230) was used to select stable cells. The survival cells were expanded and confirmed by western blot.

**Plasmid construction.** The open reading frames (ORF) of zebrafish *bmp8a*, mouse (*Alk2, Alk3, Alk4, Alk5, Alk7, Acvr2a, Acvr2b, Bmpr2, Tgfβr2, Ppary*) were cloned into the pCMV-C-Flag vector for the eukaryotic expression, respectively. The mouse Bmp8a and p65 ORF were cloned into the pCMV-C-HA vector for the eukaryotic expression, respectively. The zebrafish bmp8a and mouse Bmp8a ORF were cloned to pLVX-mCMV-ZsGreen1-Puro vector, respectively. Dominant negative mutant plasmids Alk3-ΔGS (delete the sequence of 5'-AGCTCTGG-GAGTGGATCTGGA-3' encoding the GS domain), Alk4-ΔGS (delete the sequence of 5'-TCAGGGTCTGGCTCAGGG-3' encoding the GS domain), Alk5-ΔGS (delete the sequence of 5'-TCAGGGTCTGGATCAGGT-3 ' encoding the GS domain) were cloned to pLVX-mCMV-ZsGreen1-Puro vector, respectively. The sequences (shRNA-*Bmp8a*#1: ACACCGTAACATGGTGGTCAA; shRNA-*Bmp8a*#2: ACAGCCTTTCATGGTAACCTT; shRNA-scrambled: CCTAAGGT-TAAGTCGCCCTCG) were cloned to pLVX-shRNA2-Puro vector, respectively. The deletions and mutations were created using Mut Express II Fast Mutagenesis Kit V2 (Vazyme, #C214-01). All constructs were confirmed by DNA sequencing. The primer sequences used in this study were listed in Supplemental Table 1.

**Weight measurement.** Weight (g) was measured by putting the fish into a small beaker of facility water on a scale and subtracting the non-fish weight.

**Anesthesia.** Tricaine (Sigma-Aldrich, #E10521) was used at 0.02% in water, at 28.5 °C. Fish were transferred to a beaker containing Tricaine and then monitored for signs that they had reached loss of reactivity. For anesthetics, loss of reactivity was typically reached within 60 s[58].

**Hepatic histopathology analysis.** The tissue was fixed, dehydrated, embedded and Oil red stained according to procedures[59]. In brief, fresh livers were washed two times with PBS to remove impurities such as blood and then fixed in 4% (w/v) paraformaldehyde (PFA, Beyotime, #P0099) for 2 h at 4 °C. Tissue blocks were embedded in opti-mum cutting temperature compound (OCT, Leica, #03803389). The 5 µm thick liver sections were prepared using Cryostat Microtome (Leica, CM1950). Finally, samples were subjected to Oil Red O (Solarbio, #G1260) staining. The histopathological examination of the liver was performed under a Zeiss Axio Imager A1 microscope.

**Determination of serum levels of TC, TG, and GLU.** The zebrafish were fasted for 12 h and then anesthetized. Blood was collected by holding a heparinized microcapillary tube (Kimble, #41B2501) after decapitation. The blood was allowed to settle for 1.5 h and then centrifuged at 4 °C, 12,000 × g, for 12 min. The whole blood levels of TG, TC, and GLU were measured by a fully automated biochemistry analyzer (SMT-120VP, Seamaty).

**Determination of liver and adipose tissue levels of TC, TG.** The zebrafish were fasted for 12 h and then anesthetized. Fresh liver and adipose tissues were collected, rinsed with PBS (pH 7.4) at 4 °C, blotted to filter paper, weighed, placed into a homogenization vessel, homogenized by adding Isopropanol at a ratio of weight (g): Volume (ml) = 1:9 at 4 °C, centrifuged at 10,000 × g for 10 min at 4 °C, and the supernatant placed on ice to be tested. The levels of TG and TC were analyzed using assay kits (Triglyceride Colorimetric Assay Kit, #E-BC-K261-M; Total Cholesterol Colorimetric Assay Kit, #E-BC-K109-M) purchased from Elabscience Biotechnology according to the manufacturer's instructions.

**Immunoblot analysis and co-immunoprecipitation.** Cultured cells were lysed in NP-40 buffer (Beyotime, #P0013F) for 20 min on ice. After centrifugation for 10 min at 13,000 × g, 4 °C, supernatants were incubated with Protein A + G Agarose (Beyotime, #P2055) coupled to indicated antibody for overnight. The sepharose beads were washed three times with 1 ml NP-40 buffer. For immunoblot analysis, immunoprecipitates or whole-cell lysates were separated by SDS-PAGE, electro-transferred to PVDF membranes and blocked for 4 h with 4% bovine serum albumin (BSA) in PBS-T (phosphate buffered saline supplement with 0.1% Tween 20), followed by blotting with the appropriate antibodies and detection by Omni-ECL™Femto Light Chemiluminescence Kit (Epizyme, #SQ201). The membrane was visualized using a fluorescent Western blot imaging system (ChampChemi™ 610

plus, Sagecreation). The integrated absorbance (IA = mean grey value) of the protein bands was measured using ImageJ. The target protein expression level was presented as the ratio of the IA of the target protein to the IA of β-actin or total protein.

**Antibodies**. Antibodies from Affinity Biosciences: p-Smad1/5/8 (1:1000, #AF8313), Smad1/5/8 (1:1000, #AF0614), PPARγ (1:1000, #bs-4590R), UCP1 (1:1000, #AF519). Antibodies from Beyotime: HA tag (1:1000, #AH158), Flag tag (1:1000, #72298). Antibodies from Sangon Biotech: p-ACC (1:1000, #D155180), ACC (1:1000, #D155300). Antibodies from Abcam: BMP8A (1:1000, #154373). Antibodies from ZenBio: p-AMPK (1:1000, #R26252), AMPK (1:1000, #380431), PGC-1α (1:1000, #381615). Antibodies from CWBIO: goat anti-Rabbit IgG HRP secondary antibody (1:4000, #CW0103S), goat anti-Mouse IgG HRP Conjugated (1:4000, # CW0102). Antibodies from Bioss: p38 MAPK (1:1000, #bs-0637R), p-p38 MAPK (1:1000, #bs-2210R), β-actin (1:2000, #bs-0061R), p-JNK (1:1000, #bs-1640R), JNK (1:1000, #bs-2592R), C/EBPα (1:1000, #AF6333), p-Smad2/3 (1:1000, #AF3367), Smad2/3 (1:1000, #AF6367), p-ERK1/2 (1:1000, #AF1015), ERK1/2 (1:1000, #AF0155), p-p65 (1:1000, #AF2006), p65 (1:1000, #AF5006), p-IKKα/β (1:1000, #AF3013), IKKα/β (1:1000, # AF6014).

**Drug treatment and analysis**. Smad1/5/8 inhibitor DMH1(Selleck, #S7146), Smad2/3 inhibitor TP0427736 HCl (Selleck, #S8700), were dissolved in dimethyl sulfoxide (DMSO). The cells were treated with each inhibitor, which was diluted with culture medium at concentrations of 5 μM.

**In vivo labeling and imaging of zebrafish adipocytes**. Nile Red (Solarbio, # N8440) was dissolved in acetone at 1.25 mg/ml and stored in the dark at 20 °C. Vessels containing live unanesthetized zebrafish were supplemented with Nile Red to a final working concentration of 0.5 μg/ml and then placed in the dark for 30 min. Zebrafish were anesthetized in Tricaine (Sigma-Aldrich, #E10521), mounted in 3% methylcellulose, and imaged using a Leica MZ16F fluorescence stereomicroscope.

**Luciferase assays**. Luciferase activity levels were measured according to the manufacturer's instructions (Luc-Pair Duo-Luciferase Assay Kits 2.0, iGene Biotechnology, #LF002). Briefly, HEK293T cells were plated at $6 \times 10^4$ cell/well in 24-well plates and cotransfected with various constructs at a ratio of 10:10:1 (BRE- or CAGA-driven luciferase reporter/expression plasmid/pRL-TK) using FuGENE HD Transfection Reagent. The luciferase reporter activity was measured using the Spark 20 M multifunctional microplate reader. Data were normalized by calculating the ratio of Firefly/*Renilla* luciferase activity.

**Oil red O staining**. After induction of adipogenic differentiation, 3T3-L1 cells were rinsed three times with PBS and then fixed for 20 min with 4% PFA. The cells were treated with 60% isopropanol in $H_2O$ for 2 min and then stained in freshly diluted Oil Red O solution (Oil Red O Saturated Solution (Solarbio, #G1260) was diluted with water (3:2), filtered through a 0.45 μm filter) for 30 min. Cells were then washed with 60% isopropanol in $H_2O$ and twice with PBS. The cells were observed in $H_2O$ under a microscope and photographed. Oil Red O was extracted with 100% isopropanol, and the absorbance reading was performed at $OD_{492}$ nm.

**RNA quantification**. RNA was isolated using the Trizol Reagent (Invitrogen, #15596026). Generally, RNA was reversed to cDNAs by PrimeScript™ RT reagent Kit with gDNA Eraser (TaKaRa, #RP047A). Samples without reverse transcriptase were also added as control. Gene expression was determined by amplifying the cDNA with ChamQ SYBR Color qPCR Master Mix (Vazyme, #Q431-02) by using an ABI 7500 Fast Real-Time PCR System (Applied Biosystems, USA). Gene expression levels were normalized to an internal control gene (zebrafish *β-actin* or mouse *Gapdh*). All qRT-PCR experiments were performed in triplicate and repeated three times. The primer sequences were described in Supplementary Table 2.

**RNA sequencing**. RNA sequencing was accomplished by Beijing Baimaike Biotechnology Co., Ltd (Beijing, China). Total RNA was isolated from 3T3-L1 cells using the TRIzol reagent (Invitrogen, #15596026). For RNA-sequencing analysis, three independent samples from each group, including (group 1, LV-ZsGreen1), (group 2, LV-*bmp8a*), and (group 3, LV-Bmp8a), were collected. Sequencing libraries were generated using the NEBNext Ultra Directional RNA Library Prep Kit for Illumina (NEB, #E7530L). Sequencing was performed on an Illumina NovaSeq 6000, and 150-nucleotide paired-end reads were generated. At least 6 GB of clean data with >94% of them above Q30 were produced for each sample. HISAT2 and StringTie were used to align the reads and to analyze the transcripts[60,61]. The DEGseq R package was used to identify differentially expressed genes. The whole analysis was performed on BMKCloud (www.biocloud.net).

**Statistics and reproducibility**. All the experiments were performed in triplicate and three independent repeats. Statistical analyses were performed using GraphPad Prism 8.0.2. Data are presented as mean ± SD. One-way ANOVA or two-tailed

Student's *t*-test was used to determine the *p* value. *$p < 0.05$, **$p < 0.01$, ***$p < 0.001$; ns, not significant, $p > 0.05$.

**Reporting summary**. Further information on research design is available in the Nature Portfolio Reporting Summary linked to this article.

## Data availability

The RNA-seq data have been deposited in NCBI Gene Expression Omnibus (GEO) under the accession number GSE233566. The source data for the graphs are provided as an Excel file in Supplementary Data 1. Supplementary Fig. 6–11 contain uncropped and unedited blot/gel images with size marker. All data are available from the corresponding author upon reasonable request.

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

## Acknowledgements

This work was supported by the grants of Marine S&T Fund of Shandong Province for Pilot National Laboratory for Marine Science and Technology (Qingdao) (No.2022QNLM050101-2, was updated as Science & Technology Innovation Project of Laoshan Laboratory No. LSKJ202203002), Shandong Provincial Natural Science Foundation (ZR2022MC032 and ZR2020MC050) and China Postdoctoral Science Foundation (2022M712995).

## Author contributions

S.-J.Z., Z.-H. L., and C. S. conceived and coordinated the project. S.-J.Z. and Z.-H. L. designed the experiments. S.-J.Z. and L.-H.C performed experiments and analyzed the data. X.-Y. L., X.-Y. W., and G.-D. J. contributed to the 3T3-L1 cells experiments. S.-J.Z., Z.-H.L., and C. S. wrote the manuscript, with input from the other authors. All authors reviewed the manuscript and approved the final version.

## Competing interests

The authors declare no competing interests.
