## [Peer Review File · Communications Biology]

Reviewers' comments:

Reviewer #1 (Remarks to the Author):

The manuscript by Zhong et al. comprehensively explore the role of Bmp8a in adipose tissue by using zebrafish as a model. The study is remarkably complete and provides many mechanistic observations supporting the role of Bmp8a in limiting adipogenesis and promoting fatty acid oxidation. More precisely, starting from the obese phenotype of Bmp8a mutant zebrafish, authors highlight the action of Bmp8a in regulating fatty acid oxidation through phosphorylation of AMPK and ACC. In addition, they show Bmp8a ability to inhibit PPARg transcription through Smad3/3 binding to PPARg promoter. Finally, they also highlight the action of Bmp8a on NfKb signaling, which interferes with PPARg transcriptional activity on its target genes. Authors present a good amount of data supporting their conclusions, obtained with complementary methods. Given the few available information about the role of Bmp8a, the work is adding new interesting pieces to the puzzle of adipose tissue physiology regulation. However, some concerns is raised and should be addressed to clarify some of the conclusions.

Major comments:

1. Considering the action of Bmp8a on inflammatory signaling pathways, such as NfKb, it would be important to check what is the inflammatory status of adipose tissue and liver of mutant bmp8a $-/-$ zebrafish. Is the Bmp8a effect on inflammation highlighted in cells found back in a more complex system, such as the mutant? This is very important as induction of adipogenesis is not necessarily bad. Indeed, when adipose tissue function is ok, a good storage of lipids in adipose tissue might be beneficial and problems starts when inflammation also occurs. Given the interesting effect of Bmp8a not only on adipogenesis, but also on inflammation, this aspect needs to be better addressed and also better discussed.
2. Author's observations in cells suggest a role of Bmp8a in inhibiting PPARg expression, and the subsequent expression of PPARg target genes. It is not clear how do authors reconcile these data with the reduced expression of many PPARg target genes, such as adiponectin and lpl that they observe in obese mutant zebrafish. How is the expression on PPARg in mutants Bmp8a $-/-$?
3. In general the manuscript would benefit a lot from an extensive English correction, as there are quite a lot of sentence with mistakes in verbal forms and reader has to guess a bit what authors want to say.

Minor points:

4. At what extent is Bmp8a temporal expression overlapping with that of PPARg and CEBPa along 3T3 differentiation? It would be nice to add such expression lines in figure 3b, as with data normalized at 1 in each mock at each time point in figure 3g-j, we cannot have this vision.
5. At what extent genes whose expression is significantly altered in cells transfected with mouse and zebrafish bmp8a overlap? Is there a good number of common genes in the two groups?
6. It is not clear if RNA-seq data have been deposited in GEO or similar repositories.
7. It was not possible to have access to source data.

Reviewer #2 (Remarks to the Author):

This manuscript examines the effects of BMP8a on metabolism using a bmp8a null zebrafish model, as well as 3T3-L1 and ZFL cells. The data show that anti-fat effect of Bmp8a is mediated by promoting fatty acid oxidation and inhibiting adipocyte differentiation.

Overall, this manuscript brings interesting information, and the experimental settings are solid. From a discussion point of view, it would be interesting to comment about the possible central actions of BMP8a, a mechanism through which other BMPs (for example, BMP8B) exert important actions on energy homeostasis.

Point-by-Point Responses to the Reviewers' Critiques (COMMSBIO-23-1011)

We deeply appreciate the thorough analysis and constructive suggestions provided by the two reviewers to guide us to further improve our manuscript. As described in more detail below, we have addressed all the reviewers' concerns. With this extensive revision, we hope that the reviewers will concur with us that we have addressed all of the raised concerns in a satisfactory manner and, consequently, substantially strengthened our paper.

Reviewer #1 (Remarks to the Author):

The manuscript by Zhong et al. comprehensively explore the role of Bmp8a in adipose tissue by using zebrafish as a model. The study is remarkably complete and provides many mechanistic observations supporting the role of Bmp8a in limiting adipogenesis and promoting fatty acid oxidation. More precisely, starting from the obese phenotype of Bmp8a mutant zebrafish, authors highlight the action of Bmp8a in regulating fatty acid oxidation through phosphorylation of AMPK and ACC. In addition, they show Bmp8a ability to inhibit PPAR γ transcription through Smad2/3 binding to PPAR γ promoter. Finally, they also highlight the action of Bmp8a on NfKb signaling, which interferes with PPAR γ transcriptional activity on its target genes. Authors present a good amount of data supporting their conclusions, obtained with complementary methods. Given the few available information about the role of Bmp8a, the work is adding new interesting pieces to the puzzle of adipose tissue physiology regulation.

Response: We greatly appreciate your careful review and positive comments on our work.

1. *Considering the action of Bmp8a on inflammatory signaling pathways, such as NFKb, it would be important to check what is the inflammatory status of adipose tissue and liver of mutant $bmp8a^{-/-}$ zebrafish. Is the Bmp8A effect on inflammation highlighted in cells found back in a more complex system, such as the mutant? This is very important as induction of adipogenesis is not necessarily bad. Indeed, when adipose tissue function is ok, a good storage of lipids in adipose tissue might be beneficial and problems starts when inflammation also occurs. Given the interesting effect of Bmp8a not only on adipogenesis, but also on inflammation, this aspect needs to be better addressed and also better discussed.*

Response: We sincerely appreciate the valuable and professional comments. We totally agree that induction of adipogenesis is not necessarily bad. We also agree that it would be important to check what is the inflammatory status of adipose tissue and liver of mutant $bmp8a^{-/-}$ zebrafish. Following this suggestion, we added the experiment. It was found that the expression of pro-inflammatory genes (*tnfa* and *il-1 β*) were significantly elevated in $bmp8a^{-/-}$ zebrafish liver and adipose tissue (Figure R1). This is in line with our observation by Oil Red O staining on the liver sections that a prominent fatty liver appeared in $bmp8a^{-/-}$ zebrafish (Fig. 1e). Generally, pro-inflammatory cytokines are produced in obesity (Engin et al., 2019). Another question is how to explain the finding that Bmp8a inhibits the preadipocyte differentiation into adipocyte by activating the NF- κ B inflammatory signaling pathway in 3T3-

L1 cells. From our view, they are two different biological processes: the experiment in 3T3-L1 cells is about the preadipocyte differentiation, while the experiment in mutant *bmp8a*^{-/-} zebrafish is about the adipose biology (inflammatory status) in obesity. Our data did provide evidence on the interesting effect of Bmp8a not only on adipogenesis, but also on inflammation. We added the figure and information to the discussion part in the revised manuscript to help readers understand this field comprehensively.

Figure R1: Expression of *tnfa* and *il-1β* in the WT and *bmp8a*^{-/-} zebrafish. a. b The qPCR analysis of *tnfa* and *il-1β* mRNA level in the liver (a, *n* = 3) and adipose tissue (b, *n* = 3) from WT or *bmp8a*^{-/-} zebrafish. Data were analyzed by Student's *t*-test (two-tailed). All data were presented as mean ± SD (**p* < 0.05, ***p* < 0.01, ****p* < 0.001).

2. Author's observations in cells suggest a role of Bmp8a in inhibiting PPARγ expression, and the subsequent expression of PPARγ target genes. It is not clear how do authors reconcile these data with the reduced expression of many PPARγ target genes, such as adiponectin and *lpl* that they observe in obese mutant zebrafish. How is the expression on PPARγ in mutants Bmp8a^{-/-} ?

Response: We sincerely appreciate the valuable and professional questions. Following the suggestion, we checked the expression of *ppary* in mutants *bmp8a*^{-/-} zebrafish. It was found that the expression of *ppary* was increased in both liver and adipose tissue of *bmp8a*^{-/-} zebrafish compared to wild-type zebrafish (Figure R2). This does not seem reconcile with the reduced expression of *adiponectin* and *lpl* in obese mutant zebrafish. We tend to explain from the following aspects: a) The expression of *lpl* and *adiponectin* may also be affected by genes other than *ppary* *in vivo*. b) The data from 3T3-L1 cells (*in vitro* experiment) are not exactly corresponding to the data from obese mutant zebrafish (*in vivo* experiment), because they are two different biological processes: *in vitro* experiment is about the preadipocyte differentiation, while *in vivo* experiment is about the adipose biology (inflammatory status) in obesity. There are other similar reports. For example, it was found that the adiponectin level in obese patients was significantly lower than in non-obese people (Piestrzeniewicz et al., 2007; Okrzeja et al., 2022). Anyway, this is an interesting issue and should be considered carefully. We added the figure and information to the discussion part in the revised manuscript.

Figure R2: Expression of *ppary* in the WT and *bmp8a*^{-/-} zebrafish. a. b The qPCR analysis of *ppary* mRNA level in the liver (a, $n = 3$) and adipose tissue (b, $n = 3$) from WT or *bmp8a*^{-/-} zebrafish. Data were analyzed by Student's *t*-test (two-tailed). All data were presented as mean \pm SD (** $p < 0.01$, *** $p < 0.001$).

3. In general the manuscript would benefit a lot from an extensive English correction, as there are quite a lot of sentence with mistakes in verbal forms and reader has to guess a bit what authors want to say.

Response: Thanks for the comments. We have carefully reviewed the manuscript and corrected the grammatical and typing errors as much as possible according to the suggestion in the revised manuscript. Moreover, we invited Dr. Yujun Liang, who had worked at Yale University, to correct our language.

4. At what extent is *Bmp8a* temporal expression overlapping with that of *PPARg* and *CEBPa* along 3T3 differentiation? It would be nice to add such expression lines in figure 3b, as with data normalized at 1 in each mock at each time point in figure 3g-j, we cannot have this vision.

Response: This is a good suggestion. As suggested, the *Pparg* and *C/ebpα* expression pattern during 3T3-L1 cells differentiated into adipocytes has been added in Fig.3b in the revised manuscript.

5. At what extent genes whose expression is significantly altered in cells transfected with mouse and zebrafish *bmp8a* overlap? Is there a good number of common genes in the two groups?

Response: This is a good question. Venn plot was used to identify genes that share a common expression pattern. There were 870 key molecules derived from the overlapping differentially expressed genes, including 536 upregulated genes (Figure R3) and 334 downregulated genes (Figure R4), in cells transfected with mouse and zebrafish *bmp8a*. We added the figures and information in the revised manuscript.

up-regulated genes

Figure R3: Venn diagram of up-regulated genes expression in two compared groups.

down-regulated genes

Figure R4: Venn diagram of down-regulated genes expression in two compared groups.

6. *It is not clear if RNA-seq data have been deposited in GEO or similar repositories.*

Response: RNA-seq data have been deposited in GEO (GEO accession: GSE233566).

7. *It was not possible to have access to source data.*

Response: Source data have been uploaded as supplementary material.

Reviewer #2 (Remarks to the Author):

This manuscript examines the effects of BMP8a on metabolism using a bmp8a null zebrafish model, as well as 3T3-L1 and ZFL cells. The data show that anti-fat effect of Bmp8a is mediated by promoting fatty acid oxidation and inhibiting adipocyte differentiation. Overall, this manuscript brings interesting information, and the experimental settings are solid. From a discussion point of view, it would be interesting to comment about the possible central actions of BMP8a, a mechanism through which other BMPs (for example, BMP8B) exert important actions on energy homeostasis.

Response: We thank the reviewer for the encouraging words and the appreciation of our study. We also thank the reviewer for the important suggestions. We added the discussion on the possible central actions of BMP8a, as well as BMP8B, in the revised manuscript.

REVIEWERS' COMMENTS:

Reviewer #1 (Remarks to the Author):

Authors have adequately addressed my comments.